

# Quantifying upper limb motor impairment in people with Parkinson's disease: a physiological profiling approach

Lewis A. Ingram[1,2], Vincent K. Carroll[3,4], Annie A. Butler[1,2], Matthew A. Brodie[1,2], Simon C. Gandevia[1,2] and Stephen R. Lord[1,2]

[1] Neuroscience Research Australia, Sydney, New South Wales, Australia
[2] University of New South Wales, Sydney, New South Wales, Australia
[3] NSW Health, Mid North Coast Local Health District, Coffs Harbour, New South Wales, Australia
[4] Parkinson's NSW, Sydney, New South Wales, Australia

## ABSTRACT

**Background**. Upper limb motor impairments, such as slowness of movement and difficulties executing sequential tasks, are common in people with Parkinson's disease (PD).

**Objective**. To evaluate the validity of the upper limb Physiological Profile Assessment (PPA) as a standard clinical assessment battery in people with PD, by determining whether the tests, which encompass muscle strength, dexterity, arm stability, position sense, skin sensation and bimanual coordination can (a) distinguish people with PD from healthy controls, (b) detect differences in upper limb test domains between "off" and "on" anti-Parkinson medication states and (c) correlate with a validated measure of upper limb function.

**Methods**. Thirty-four participants with PD and 68 healthy controls completed the upper limb PPA tests within a single session.

**Results**. People with PD exhibited impaired performance across most test domains. Based on validity, reliability and feasibility, six tests (handgrip strength, finger-press reaction time, 9-hole peg test, bimanual pole test, arm stability, and shirt buttoning) were identified as key tests for the assessment of upper limb function in people with PD.

**Conclusions**. The upper limb PPA provides a valid, quick and simple means of quantifying specific upper limb impairments in people with PD. These findings indicate clinical assessments should prioritise tests of muscle strength, unilateral movement and dexterity, bimanual coordination, arm stability and functional tasks in people with PD as these domains are the most commonly and significantly impaired.

Corresponding authors
Lewis A. Ingram,
l.ingram@neura.edu.au,
l.ingram@neura.edu.au
Stephen R. Lord, s.lord@neura.edu.au

# INTRODUCTION

Parkinson's disease (PD) is a progressive, degenerative neurological condition characterised by rigidity, slowness of movement, resting tremor and postural instability (*Kalia & Lang, 2015*; *Jankovic, 2008*), the degree to which varies considerably from patient to patient

and changes over time (*Jankovic, 2008*; *Politis et al., 2010*). These motor impairments result from the destruction of neurons in the substantia nigra, a region of the midbrain responsible for the production of dopamine (*Schneider & Obeso, 2015*; *Obeso et al., 2008*; *Kordower et al., 2013*), a neurotransmitter that is critical for the facilitation of voluntary movement (*Harrington & Haal, 1991*). This loss in dopaminergic cells is thought to inhibit the motor thalamic nuclei, resulting in decreased excitation at the cortical level (*Burciu & Vaillancourt, 2018*; *Broski et al., 2014*; *Albin, Young & Penney, 1989*; *Graybiel, 1990*). In the upper extremities, these pathological changes can manifest as difficulties in reaching, grasping and manipulating objects with sufficient speed and dexterity (*Proud & Morris, 2010*). Such upper limb impairments commonly lead to difficulties with eating, drinking and self-care and can restrict participation in work, recreation and social activities (*Peto et al., 1995*; *Morris, 2000*).

While clinical guidelines are provided for the assessment of balance and gait in people with PD, guidelines for clinical assessments of upper limb functioning across multiple domains are comparatively less developed (*Aragon, Ramaswamy & Ferguson, 2007*; *Keus et al., 2004*). Indeed, *Proud et al. (2015)* highlighted in their recent systematic review the lack of high-quality studies supporting the validity and responsiveness of the many clinical tests currently used in the assessment of upper limb function in people with PD. As a consequence, clinicians may rely on non-standardised or generic tests of manual dexterity, along with assessment tools designed for other clinical populations (*Proud et al., 2013*). Implementation of a standardised approach to quantify upper limb impairment and treatment effects would help to promote evidence-based practice for the health care needs of people with PD (*Proud et al., 2015*; *Fawcett, 2007*).

The upper limb Physiological Profile Assessment (PPA) provides quantitative measures of key physiological domains essential for upper limb function, including muscle strength, dexterity, arm stability, position sense, skin sensation and bimanual coordination (*Ingram et al., 2019*). Modelled on the original PPA (*Lord, Menz & Tiedemann, 2003*; *Lord, Delbaere & Gandevia, 2016*) as a means of quantifying fall risk in older people and neurological populations, the upper limb PPA is designed to be clinically valid, simple and quick to administer, and capable of generating quantitative scores of key domains required for adequate upper limb function. Therefore, the upper limb PPA offers an 'impairment-based' profile to complement medically-based diagnoses. Given the inter-individual variability in both the motor symptoms expressed and the relative magnitude of each, the upper limb PPA offers a means of individualising treatment and rehabilitation strategies based on a patient's specific impairments.

An additional consideration for the assessment of physical functioning in people with PD is the timing of the assessment and medication intake. In levodopa-treated patients, the 'on-off' phenomenon—characterised by a decline of motor function or, in more severe cases, sudden and unpredictable motor fluctuations, commonly occurs during the 'off' phase towards the end-of-dose (*Bhidayasiri & Truong, 2008*). To assess the extent of this change, assessment of upper limb function is required during both the 'on' state—about 30 min after the administration of levodopa, as well during the 'wearing off' period approximately 45–60 min prior to the next scheduled dose.

The aims of this validation study were to: (a) compare upper limb PPA test scores in people with PD to that of age- and gender-matched controls without PD, (b) assess associations between performance in each test and age, disease duration, disease severity and upper limb function, (c) compare test performance between the 'on' and 'off' states in people with PD, (d) to identify key tests from the upper limb PPA for assessing upper limb function in people with PD and (e) present PD group and individual upper limb physiological profiles based on established normative values for each test. We hypothesised: people with PD would demonstrate impaired performance across most upper limb PPA tests when compared to age- and gender-matched controls without PD, and within the PD group most upper limb PPA tests would discriminate between medication states with test performances associated with disease duration, disease severity and upper limb function.

## METHODS

### Participants

Thirty-four people with Parkinson's disease participated in the study (47 to 87 years, 22 males and 12 females). All participants provided written and verbal consent prior to participation. Participants were recruited from three Parkinson's New South Wales support groups located on the Mid North Coast, New South Wales, Australia. For inclusion, participants had to be aged 18 years or older, and have had received a clinical diagnosis of PD by a neurologist with a score of 1–4 on the Hoehn and Yahr (HY) scale. Prospective participants were excluded if they could not provide verbal or written consent, were unable to follow test instructions, or have any concurrent musculoskeletal and/neurological condition that could affect test performance. Testing took place in March 2019 and was conducted at a testing facility or within participants' homes.

For each participant with PD, two age- and gender-matched controls without PD were randomly selected from the database of a research study previously conducted at Neuroscience Research Australia (*Ingram et al., 2019*). Ethical approval was granted by the Human Research Ethics Committee, University of New South Wales (HC 15607). All assessments were conducted in accordance with the Declaration of Helsinki (2008).

### Procedure

Each PD participant was tested twice, i.e., in their 'on' and 'off' medication states. The order in which this occurred was determined by participant convenience and logistics. At the beginning of the initial assessment, participants completed the Disabilities of the Arm, Shoulder and Hand (DASH) questionnaire (*Hudak, Amadio & Bombardier, 1996*). The DASH is a 30-item questionnaire used to measure upper-extremity function (*Beaton et al., 2001*). Participants are scored on a 100-point scale, with higher values indicating increased levels of self-reported upper limb impairment. DASH scores were not calculated until after the completion of testing. Visual acuity was screened using a Logarithmic Visual Acuity Chart (SLOAN Two Sided ETDRS Format Near Point Test) calibrated for testing at 40 cm (Precision Vision, USA). This was to ensure that each participant had satisfactory vision to complete the subsequent tests (*Lovie-Kitchin, 2015*). Visual aids were permitted where

necessary. Participants then completed the upper limb tests (see below). A single examiner assessed all participants with each test session taking approximately 60 min.

The upper limb PPA consists of 13 tests (with a total of 16 test measures) classified into the physiological domains of muscle strength, unilateral movement and dexterity, position sense, skin sensation, bimanual coordination, arm stability and functional tasks. The tests have been shown to be sensitive in detecting motor impairments in people with compromised upper limb function from the general population. Age and gender normative scores are available for each test (*Ingram et al., 2019*). Each of the 13 tests are outlined briefly below, and have been described in detail previously (*Ingram et al., 2019*).

## Measurements

Muscle strength was assessed at the elbow (isometric elbow flexion strength) and the hand (handgrip strength). **Isometric elbow flexion strength** was assessed using a digital hanging scale anchored to a portable wooden platform situated beneath the chair while participants were seated with the elbow joint flexed at 90° and forearm supinated. **Handgrip strength** was assessed with Jamar+ Digital Dynamometer. The best of three trials (measured in kilograms) was recorded as the test score. Unilateral movement and dexterity were assessed via finger-press reaction time, finger tapping, 9-hole peg test, and the loop & wire test. **Finger-press reaction time** was assessed in milliseconds using a modified computer mouse containing a light stimulus embedded in the left button, which participants responded to by pressing the right button. The number of times participants could tap their index finger on a touch pad over a 10-second period provided the test score for the **finger tapping** test. The **9-hole peg test** (9-HPT) was measured (in seconds) as the time taken for the participant to pick up each peg individually, place them into each of the holes, and then return them to the molded dish (*Kellor M, 1971*; *Mathiowetz et al., 1985*). The average number of times the participant touched the copper wire maze on each of the two trials of the **loop & wire test** was recorded as the test score. **Position sense** was measured using an elbow matching test, where differences (in degrees) in aligning the index fingers was recorded using a clear acrylic sheet marked with a protractor that was vertically aligned to the participants midline, with the average error of 5 trials recorded as the test score. Tactile sensitivity, two-point discrimination and two-line discrimination were used to quantify skin sensation. For **tactile sensitivity**, von-Frey filaments were used to assess perceptual thresholds (measured in grams) to cutaneous stimuli at the hypothenar eminence. Static **two-point discrimination** was measured (in mm) using a small- (2–8 mm) and large-interval (9–20 mm) Mackinnon-Dellon Disk-Criminator applied along the mediolateral axis of the distal pad of the index finger. **Two-line discrimination** measured the smallest distance (in mm) that participants could accurately detect between two lines of 0.6 mm diameter as they progressively separated over a distance of 580 mm. Bimanual coordination was tested using the **bimanual pole test** where participants moved a screw embedded within a modified swivel stick back and forth through a maze as fast as possible by flexing and extending the wrists in a coordinated manner (measured in seconds). **Arm stability** was measured as the total path (in degrees) that the participants' wrist moved as they sat with their outstretched arm held perpendicular to the floor over a period of 30 s. Participants

completed four conditions (unweighted eyes open/closed; weighted eyes open/closed). The **shirt task** was used as a measure of functional performance. Here, participants were timed as they put on a long sleeve shirt and did up all six buttons (measured in seconds).

With the exception of handgrip strength (*Roberts et al., 2015*) and the 9-hole peg test (*Earhart et al., 2028*; *Shah et al., 2019*), none of the upper limb PPA tests have previously been validated in the PD population. However, measures of finger tapping speed and amplitude, tremor and bimanual coordination are all included in the MDS-UPDRS (*Movement Disorder Society Task Force on Rating Scales for Parkinson's Disease, 2003* ), underscoring their importance in the overall assessment of upper limb function in people with PD. Participants performed each unilateral test with their dominant hand, and were seated for all tests with the exception of the shirt task. The number of trials for each test were based on standard protocols for previously validated tests, and restricted to one or a few trials for some of the novel test to facilitate quick administration and preclude leaning effects.

## Data and statistical analysis

Continuous data are reported as mean ± SD or, if normality assumptions were violated, medians (interquartile range [IQR]). Independent $t$-tests and Mann–Whitney tests were used to contrast performances in each of the upper limb PPA tests between the PD group and the age- and gender-matched healthy controls. Correlations between test performance, age, duration of PD, HY scale and DASH scores were assessed using Pearson's $r$ and Spearman's rho, and were interpreted as follows: <0.3 - negligible, 0.3–0.5 - small, 0.5–0.7 - moderate, and >0.7 - strong. Differences in test performance between the 'off' and 'on' medication states in the PD group were analysed using paired t-tests and Wilcoxon matched pairs tests. Finally, to highlight the degree to which test performances are impaired, test scores for the PD group during the 'off' and 'on' states were converted to standardised (z) scores based on data from a reference cohort of people without PD (*Ingram et al., 2019*). By doing so, a score of zero indicates average performance for healthy controls aged 50 years and over and each unit represents one standard deviation: positive and negative scores indicate above and below average performances respectively. The reference cohort was used to compile the profiles, as it is common to our proposed companion papers addressing other neurological conditions such as stroke, and thus provides a robust way to contrast upper limb impairments among clinical groups. For this analysis, variables were $log_{10}$ transformed if normality assumptions were violated.

For position sense, two-line discrimination and the shirt task, no scores were available for one, three and four participants, respectively, who could not complete the task. With the exception of the two-line discrimination test, these participants were assigned a maximum score, which was calculated as the mean + 3SD's from data from 124 participants with neurological disorders: 34 participants with PD (the current sample), 50 stroke participants and 40 participants with multiple sclerosis. Maximum scores for these tests were calculated as 13.5° and 300 s for position sense and the shirt task, respectively. For the two-line discrimination test, these participants were assigned the maximum score available for this test: four mm. Furthermore, a small number of participants registered scores that

were beyond those maximum scores that were calculated from the same database of 124 participants with neurological disorders (i.e., the mean + 3SD's for each test). These included finger-press reaction time ($n = 1$, off state), 9-hole peg test ($n = 1$, on state), arm stability eyes closed ($n = 1$, off state; $n = 1$, on state), arm stability weight eyes open ($n = 1$, off state), arm stability weight eyes closed ($n = 2$, off state), and the shirt task ($n = 1$, off state). In each of these cases, the participant was assigned the maximum score for each of these tests, i.e., 500 ms, 180 s, 400° and 300 s, respectively.

All statistical analyses were performed using GraphPad Prism 7 software and Estimation Statistics Beta (Ho et al., 2019; Cumming & Calin-Jageman, 2016) with the level of significance set at $p < 0.05$. All data are presented in File S1.

## RESULTS

### Participant characteristics

Table 1 shows demographic, visual acuity, disease duration, PD severity (HY scale), and self-reported upper limb function measures (DASH scores) for the PD group ($n = 34$). The mean age was 68.6 years (range = 47–87) with women making up 35% ($n = 12$) of the sample. All but five participants identified as right-hand dominant. The mean logMAR score was 0.26 (corresponding to a Snellen eye chart score of 6/12 –adequate for all tests in the assessment battery). Twenty-seven participants performed the initial round of testing during their 'off' medication phase, between the hours of 0800 and 1000 with no PD medication taken after retiring to bed the night before. The median time since diagnosis was 5.0 years (IQR = 3.0–10.0) and the mean HY scale score was 2.7 (range = 1–4), suggestive of bilateral symptoms with impaired postural reflexes but physically independent. The mean DASH score of 36 out of 100 exceeds the threshold of 15, which indicates self-reported upper limb function (Kennedy et al., 2011).

Demographic, visual acuity and DASH scores for the age- and gender-matched controls are also shown in Table 1 ($n = 68$). Both the mean logMAR (0.14) and DASH scores (6.6) were lower than in the PD group, indicating better visual acuity and lower levels of self-reported upper limb function, respectively.

For the reference cohort of people without PD ($n = 176$), the mean age was 70.1 years (range = 50–96) with a 53:47 ratio of females to males (Table 1). Visual acuity and DASH scores were comparable to that of the age- and gender-matched control group.

### Comparison with age- and gender-matched controls without PD

During the 'off' state, the PD group performed worse than the healthy control group in all tests with the exception of isometric elbow flexion strength, two-point discrimination and two-line discrimination (Table 2 and Figs. 1–2). The same pattern of associations was evident in the 'on' state, with the exception that scores in bimanual pole test did not differ between the PD and healthy control participants (Figs. 3–4).

### Comparison with normative values

Figure 5 presents test results as standardised ($z$) scores for the PD group during the 'off' and 'on' states referenced to the reference cohort without PD (Ingram et al., 2019). These

Ingram et al. (2021), *PeerJ*, DOI 10.7717/peerj.10735

**Table 1** Parkinson's disease participant ($n = 34$), age- and gender-matched healthy control participant characteristics ($n = 68$) and 50+ population group ($n = 176$).

| | Age: mean (range) | Gender: F:M (ratio %) | Handedness (right) k (%) | Visual acuity (logMAR) (SD) | Order of testing: 'Off' state first (ratio %) | Years with PD: median (IQR) | Hoehn and Yahr scale: mean (range) | DASH score: median (IQR) |
|---|---|---|---|---|---|---|---|---|
| PD group | 68.6 (47–87) | 12:22 (35:65) | 29 (85) | 0.26 (0.21) | 27:7 (79:21) | 5.0 (3.0–10.0) | 2.7 (1–4) | 36.6 (18.6–48.7) |
| Control group | 68.5 (46–87) | 24:44 (35:65) | 59 (87) | 0.14 (0.21) | | | | 3.0 (0.8–8.8) |
| 50+ population | 70.1 (50–96) | 94:82 (53:47) | 161 (92) | 0.11 (0.21) | | | | 3.3 (0.8–10.8) |

Ingram et al. (2021), *PeerJ*, DOI 10.7717/peerj.10735

Peer*J*

**Table 2** Mean ± SD or median (IQR) scores for PD group during their 'off' and 'on' medication states and non-PD controls for each test, mean or median difference [95% CI] in performance between 'off' state PD and non-PD controls, and 'on' state PD and non-PD controls (independent *t*-tests or Mann–Whitney *U* tests).

| Measure | PD group (n = 34) | | Control group (n = 68) | Difference | |
|---|---|---|---|---|---|
| | 'Off' state | 'On' state | | 'Off' vs. control | 'On' vs. control |
| Isometric elbow flexion strength (kg) | 19.6 ± 8.8 | 20.9 ± 9.3 | 23.5 ± 9.8 | 3.9 [0.1, 7.7] | 2.6 [−1.2, 6.6] |
| Handgrip strength (kg) | 29.4 ± 11.1 | 29.2 ± 11.1 | 37.8 ± 12.2 | 8.4 [3.7, 12.8][**] | 8.6 [3.7, 13.1][***] |
| Finger-press reaction time (ms) | 249.3 (208.0–274.1) | 236.5 (208.3–292.7) | 206.0 (189.6–234.4) | −43.3 [−65.0, −11.0][a***] | −30.5 [−73.6, −7.7][a***] |
| Finger tapping (no. of taps) | 50.5 (42.8–55.0) | 50.5 (46.0–57.3) | 56.0 (49.0–60.0) | 5.5 [0.0, 9.0][a***] | 5.3 [1.7, 9.7][**] |
| 9-hole peg test (s) | 28.2 (24.1–35.9) | 25.0 (22.1–32.6) | 21.5 (19.5–23.7) | −6.6 [−9.5, −2.8][a***] | −3.5 [−7.4, −1.3][a***] |
| Loop & wire test (no. of touches) | 26.5 (18.6–41.4) | 23.5 (16.5–41.1) | 12.8 (7.6–19.9) | −13.8 [−20.8, −6.5][a***] | −10.8 [−21.0, −5.0][a***] |
| Position sense (°) | 4.8 (3.4–6.9) | 4.4 (2.4–5.7) | 2.6 (1.8–3.8) | −2.2 [−3.6, −1.1][a***] | −1.8 [−2.9, −0.8][a**] |
| Tactile sensitivity (g) | 0.28 (0.07–0.40) | 0.12 (0.07–0.40) | 0.07 (0.04–0.16) | −0.21 [−0.33, 0.00][a**] | −0.05 [−0.33, 0.00][a*] |
| Two-point discrimination (mm) | 3.8 ± 1.1 | 3.6 ± 1.0 | 3.6 ± 1.1 | −0.2 [−0.7, 0.3] | 0.0 [−0.4, 0.4] |
| Two-line discrimination (mm) | 2.3 (2.0–2.5) | 2.3 (1.9–2.7) | 2.2 (1.9–2.4) | −0.1 [−0.3, 0.1][a] | −0.1 [−0.4, 0.0][a] |
| Bimanual pole test (s) | 20.4 (16.9–39.5) | 19.1 (16.0–37.6) | 18.1 (14.4–25.1) | −2.3 [−10.7, 0.7][a*] | −0.9 [−11.7, 1.7][a] |
| Arm stability –Eyes open (°) | 54.5 (40.0–93.4) | 52.5 (38.6–80.1) | 39.6 (33.9–48.7) | −14.9 [−33.0, −2.3][a***] | −13.0 [−29.2, 0.9][a**] |
| Arm stability –Eyes closed (°) | 50.7 (39.5–86.9) | 52.6 (37.1–94.5) | 40.5 (33.3–48.2) | −10.2 [−24.5, −0.7][a**] | −12.2 [−28.5, 0.8][a**] |
| Arm stability –Weight eyes open (°) | 58.3 (43.2–105.6) | 55.4 (41.8–77.2) | 46.8 (38.2–56.5) | −11.4 [−44.8, 1.2][a**] | −8.6 [−24.8, 4.0][a*] |
| Arm stability –Weight eyes closed (°) | 68.9 (44.7–100.9) | 58.0 (40.9–98.3) | 46.4 (39.7–53.8) | −22.5 [−39.8, −7.3][a***] | −11.6 [−34.3, 1.7][a*] |
| Shirt task (s) | 75.1 (45.0–112.0) | 67.7 (39.0–111.8) | 32.9 (26.4–39.9) | −42.2 [−64.8, −18.3][a***] | −34.8 [−51.1, −17.9][a***] |
| DASH questionnaire | 36.6 (18.5–48.7) | | 3.0 (0.8–8.8) | −33.6 [−44.7, −25.5][a***] | |

**Notes.**

[a] Mann–Whitney *U* test performed for non-parametric variables, differences reported as median [95% CI].

[*] $p < 0.05$.

[**] $p < 0.01$.

[***] $p < 0.001$, uncorrected.

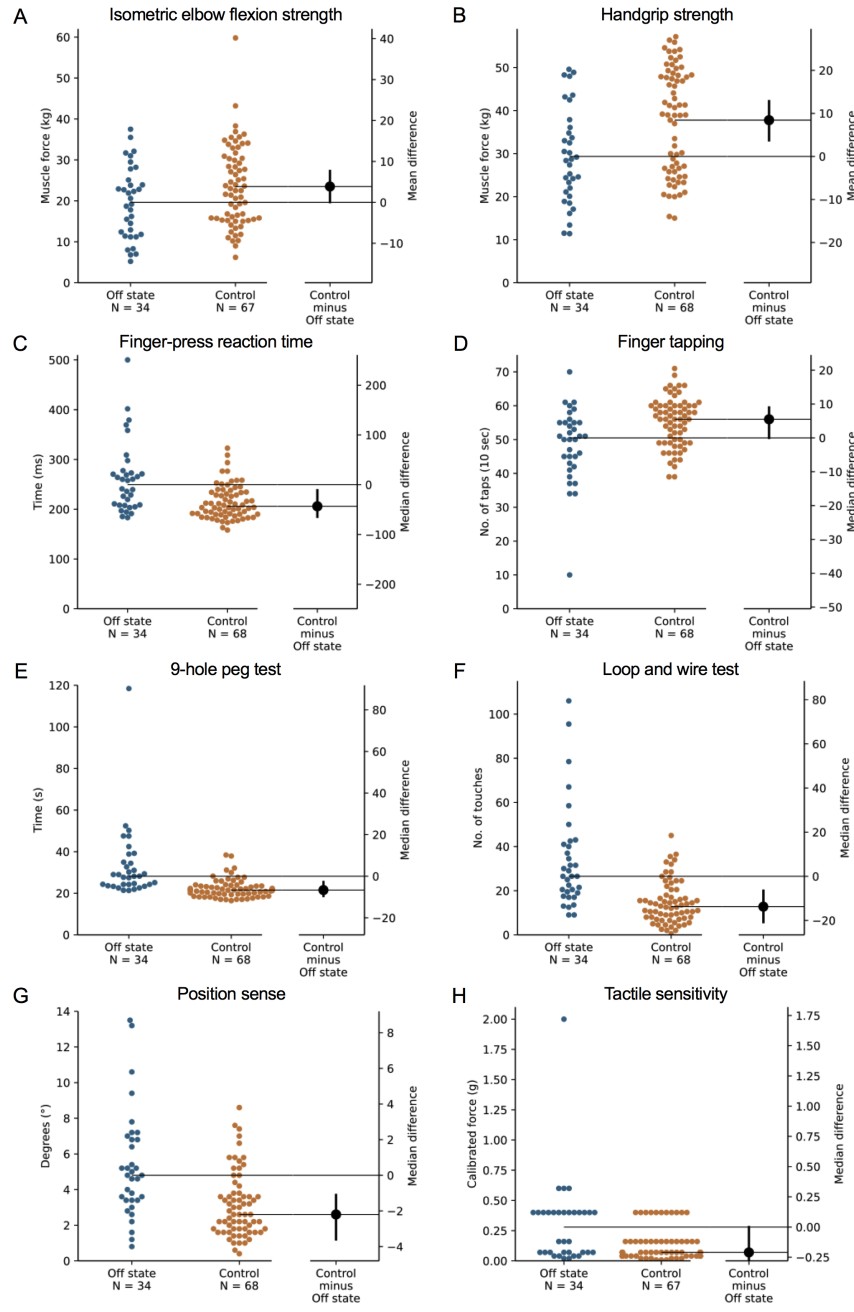

**Figure 1** Differences in performance in each test of the upper limb PPA between people with PD and age- and sex-matched healthy controls for **(A)** isometric elbow flexion strength, **(B)** handgrip strength, **(C)** finger-press reaction time, **(D)** finger tapping, **(E)** 9-hole peg test, **(F)** loop and wire test, **(G)** position sense, and **(H)** tactile sensitivity. Test scores from the PD group performed during their 'off' medication state. Blue circles represent each individual PD participant's test score for that particular test, while orange circles represent each individual control participant's test score. The black circle located along the right axis of each graph represents the mean or median difference (depending on the distribution of the test scores in each group) in test scores between the PD group and the age- and sex-matched control group. Error bars depict 95% confidence intervals. Note: due to missing data, a small number of control group test scores ($n = 1$–13) do not contain a complete set of 68 observations.

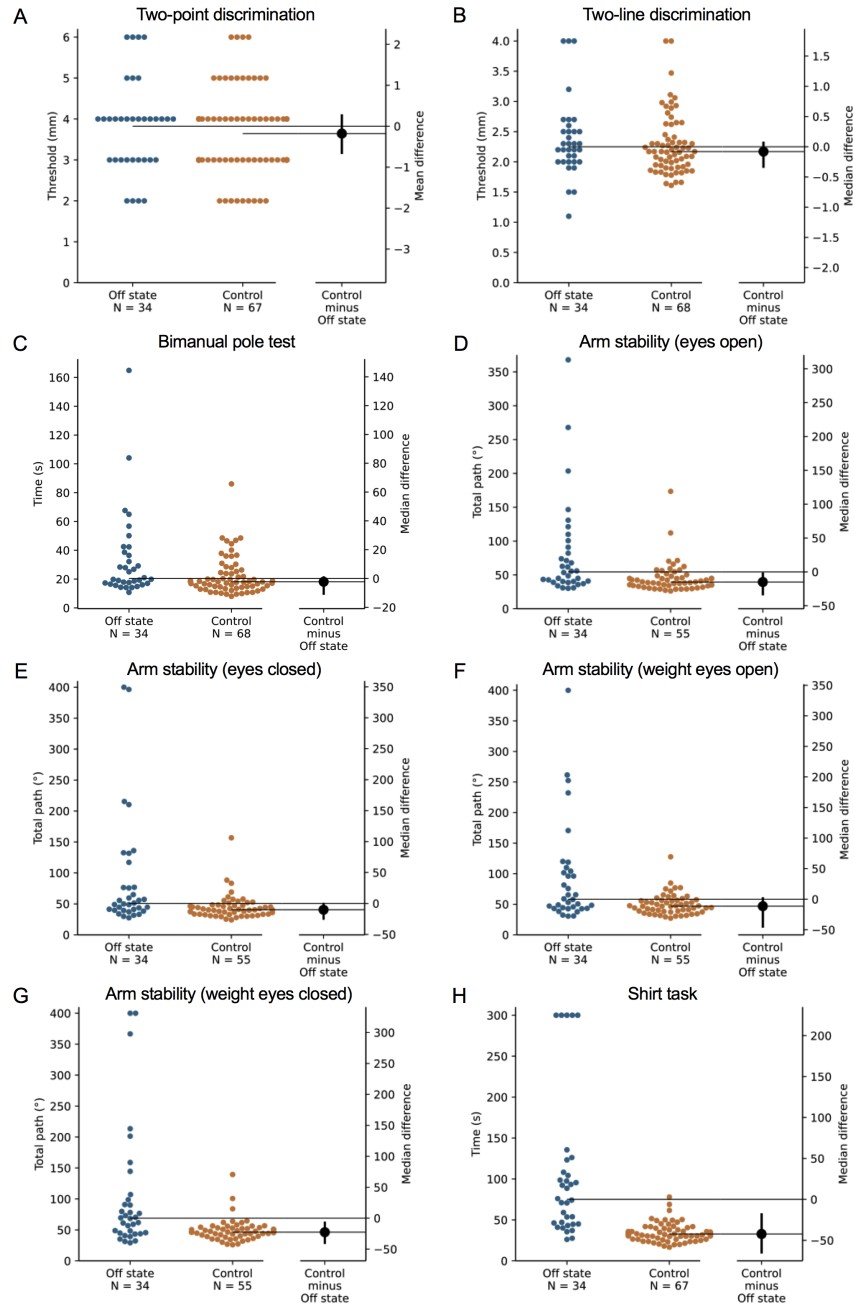

**Figure 2** Differences in performance in each test of the upper limb PPA between people with PD and age- and sex-matched healthy controls for (A) two-point discrimination, (B) two-line discrimination, (C) bimanual pole test, (D) arm stability (eyes open), (E) arm stability (eyes closed), (F) arm stability (weight eyes open), (G) arm stability (weight eyes closed), and (H) the shirt task. Test scores from the PD group performed during their 'off' medication state. Blue circles represent each individual PD participant's test score for that particular test, while orange circles represent each individual control participant's test score. The black circle located along the right axis of each graph represents the mean or median difference (depending on the distribution of the test scores in each group) in test scores between the PD group and the age- and sex-matched control group. Error bars depict 95% confidence intervals. Note: due to missing data, a small number of control group test scores ($n = 1$–13) do not contain a complete set of 68 observations.

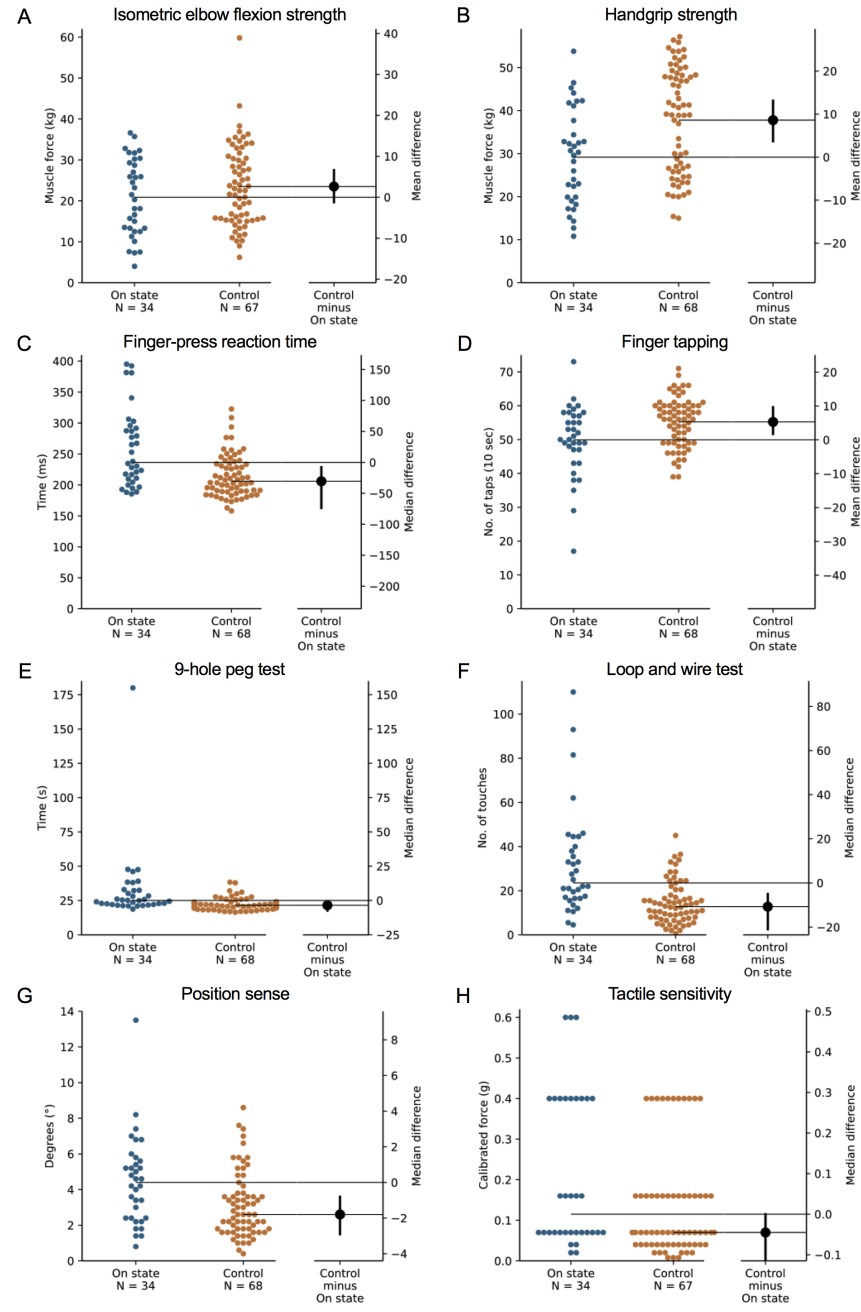

**Figure 3** Differences in performance in each test of the upper limb PPA between people with PD and age- and sex-matched healthy controls for **(A)** isometric elbow flexion strength, **(B)** handgrip strength, **(C)** finger-press reaction time, **(D)** finger tapping, **(E)** 9-hole peg test, **(F)** loop and wire test, **(G)** position sense, and **(H)** tactile sensitivity. Test scores from the PD group performed during their 'on' medication state. Blue circles represent each individual PD participant's test score for that particular test, while orange circles represent each individual control participant's test score. The black circle located along the right axis of each graph represents the mean or median difference (depending on the distribution of the test scores in each group) in test scores between the PD group and the age- and sex-matched control group. Error bars depict 95% confidence intervals. Note: due to missing data, a small number of control group test scores ($n = 1–13$) do not contain a complete set of 68 observations.

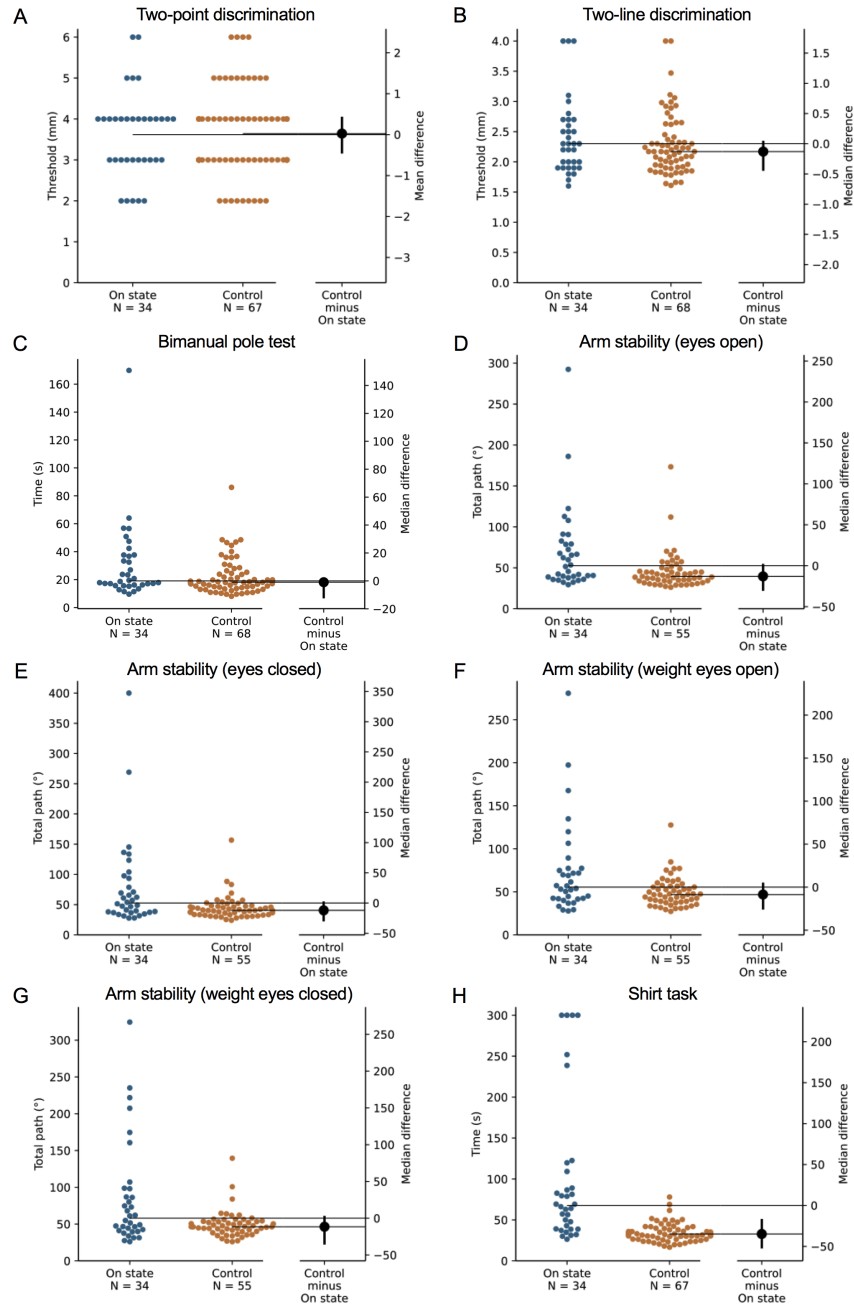

**Figure 4** Differences in performance in each test of the upper limb PPA between people with PD and age- and sex-matched healthy controls for (A) two-point discrimination, (B) two-line discrimination, (C) bimanual pole test, (D) arm stability (eyes open), (E) arm stability (eyes closed), (F) arm stability (weight eyes open), (G) arm stability (weight eyes closed), and (H) the shirt task. Test scores from the PD group performed during their 'on' medication state. Blue circles represent each individual PD participant's test score for that particular test, while orange circles represent each individual control participant's test score. The black circle located along the right axis of each graph represents the mean or median difference (depending on the distribution of the test scores in each group) in test scores between the PD group and the age- and sex-matched control group. Error bars depict 95% confidence intervals. Note: due to missing data, a small number of control group test scores ($n = 1–13$) do not contain a complete set of 68 observations.

profiles indicate that compared to healthy controls, the PD group was most impaired in the shirt task (*z* scores < −2), moderately impaired with respect to isometric elbow flexion strength (females, 'off' state only), finger-press reaction time, 9-hole peg test, loop and wire test ('off' state only), and all four conditions of the arm stability test (except for the two eyes open conditions during the 'on' state) (*z* scores between –1 and –2) and relatively less impaired with respect to the remaining tests (*z* scores > −1). Figure 6 highlights the inter-individual variability in the clinical presentation of upper limb motor impairments among four PD participants by contrasting their individual performance profiles, each derived from the reference cohort normative values.

### Test performance in relation to age, PD duration, disease severity and upper limb function

In the 'off' state, poorer performance in the handgrip strength, two-point discrimination and bimanual pole test demonstrated small associations with increased age, while during the 'on' state, age showed small and moderate associations with poorer performance in both measures of muscle strength and the 9-hole peg test (Table 3 and Figs. 7 and 8). Finger-press reaction time and tactile sensitivity were the only tests associated with time since PD diagnosis during the 'on' medication state, with the direction of the relationship indicating worse performance with longer disease duration. The strengths of these associations were small (0.3–0.5). No tests were significantly associated with disease duration during the 'off' state.

During the 'off' state, poorer performance in the 9-hole peg test, loop and wire, and bimanual pole test demonstrated small associations with increased PD severity, as measured by the HY scale, while finger-press reaction time and shirt task performance were moderately associated with increased HY scale scores. During the 'on' state, finger tapping, loop and wire, arm stability (eyes closed, weight eyes open), and shirt task performance demonstrated small associations with the HY scale, while performance in finger-tapping and 9-hole peg test were moderately associated with the HY scale.

During the 'off' state, higher levels of self-reported deficits in upper limb function, as measured by the DASH questionnaire, demonstrated small associations with reduced performance in finger-press reaction time, 9-hole peg test, bimanual pole test, and the shirt task, while both tests of muscle strength and two-point discrimination were moderately associated with the DASH questionnaire scores. Lastly, poorer performance during the 'on' state in the 9-hole peg test, and bimanual pole test, arm stability (eyes closed), and the shirt task showed small associations with higher DASH questionnaire scores, while both tests of muscle strength and finger-press reaction time were moderately associated with the DASH.

### Comparison of test scores between the 'on' and 'off' states

Mean (±SD) and median (IQR) test scores between the 'on' and 'off' medication states are presented in Table 4. Differences in performance were observed for isometric elbow flexion strength, 9-hole peg test, position sense, bimanual pole test and the two weighted conditions of the arm stability test. In each case, participants performed better during the 'on' state. There was no difference in performance between medication states for the remaining tests.

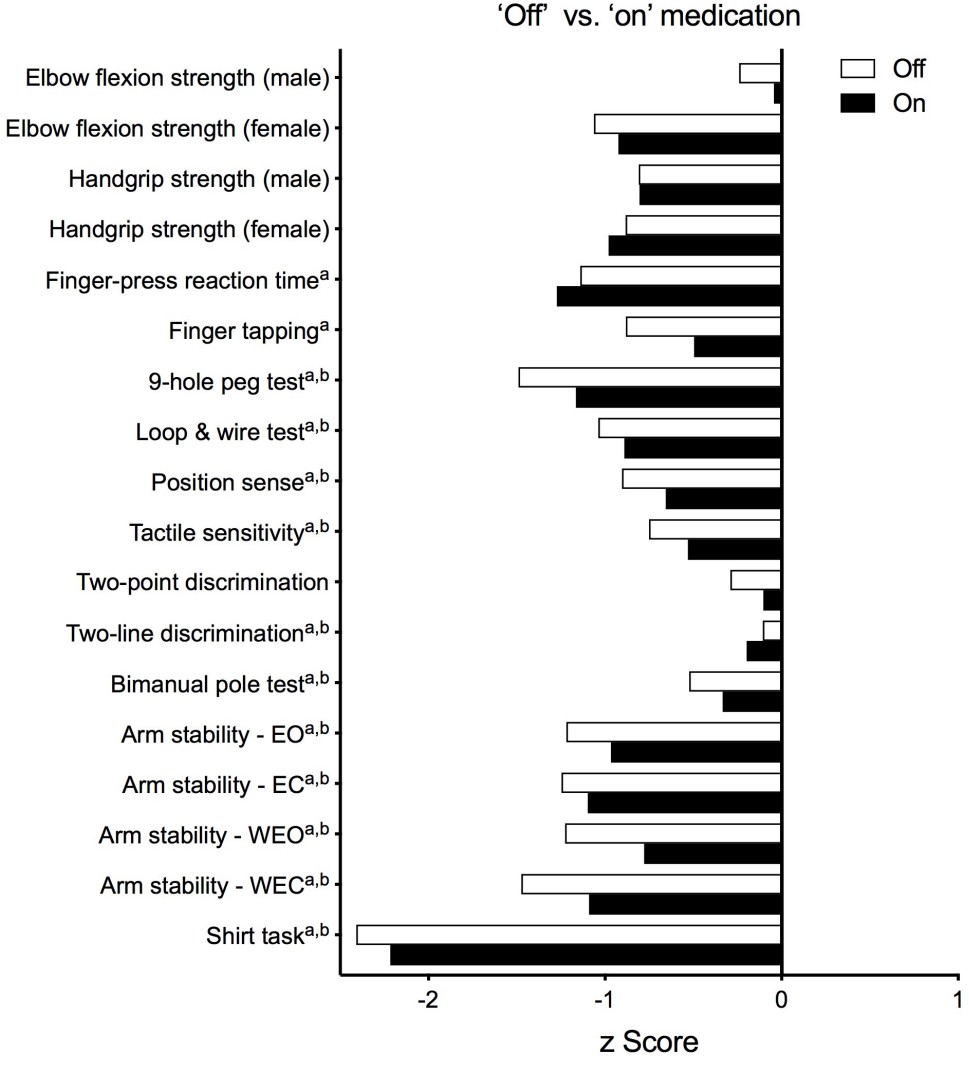

**Figure 5** Upper limb Physiological Profile Assessment (PPA) test scores presented as standardised ($z$) scores for the PD group during the 'off' and 'on' medication states referenced to a cohort of people aged 50+ years without PD. Test scores are presented as standardised ($z$) scores to allow direct comparison in performance between each test both within and between individuals. Each unit represents one standard deviation. A score of zero indicates an average level of performance compared to the reference population without PD, while negative scores represent below-average performances. Note that [a] and [b] indicate those variables that were positively skewed during the 'off' and 'on' states, respectively, and therefore converted to their logarithmic form prior to calculation of their $z$-scores. Abbreviations: EO, eyes open; EC, eyes closed; WEO, weight eyes open; WEC, weight eyes closed.

## Key tests for assessing upper limb function in people with PD

Table 5 presents a weighting system based on validity, reliability and feasibility for determining optimal tests of upper limb function in people with PD. Modelled on a scale devised to identify useful mobility tests for predicting falls in older people (*Tiedemann et al., 2008*), equal weights were allocated to (i) ability to differentiate between people with PD during their 'on' state and healthy controls, (ii) significant correlations with

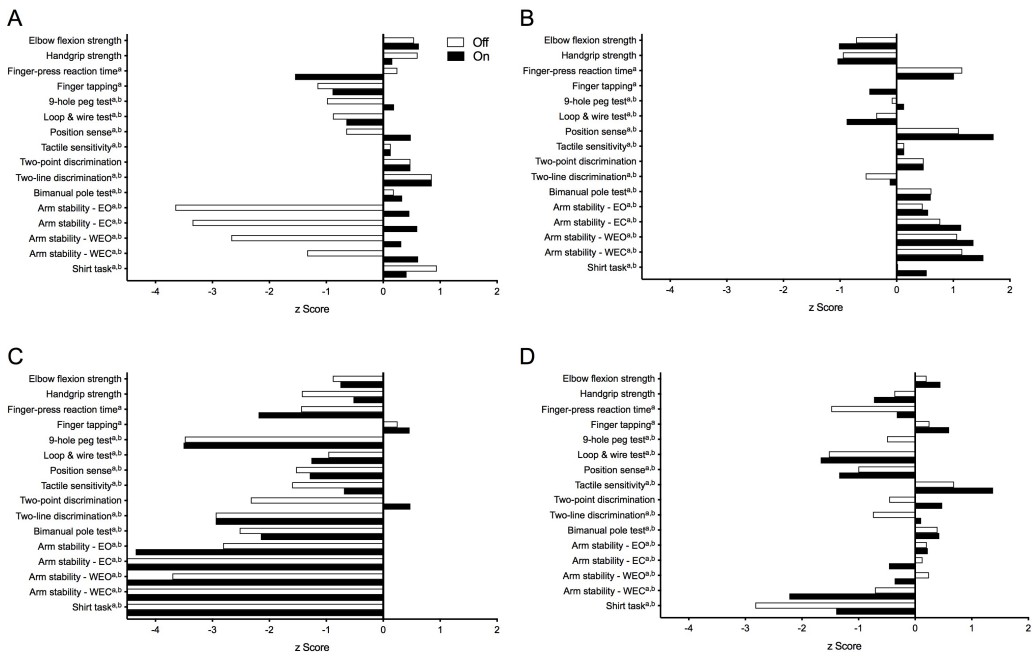

**Figure 6** **Upper limb physiological performance profiles of four individuals with PD, highlighting the inter-individual variability of upper limb motor impairment associated with PD.** Test scores are presented as standardised ($z$) scores referenced to a normative database of individuals from the general population aged 50 years and over. (A) A 58-year old male (HY = 2.0) demonstrating clear differences in performance between the 'off' and 'on' medication states in measures of arm stability. (B) A 73-year old male (HY = 1.0) who, despite below average muscle strength, displays above average levels of performance across all remaining physiological domains. (C) In contrast, a 63-year old male (HY = 3.0) who, with the exception of finger tapping speed, scores well below average in all other physiological domains. (D) A 71-year old female (HY = 3.0) with a mixed presentation, showing above average performance in measures of proximal muscle strength, motor speed and skin sensation while appearing impaired in reaction time, dexterity and functional tasks. Note: strength test scores are adjusted for gender. Note that [a] and [b] indicate those variables that were positively skewed during the 'off' and 'on' states, respectively, and therefore converted to their logarithmic form prior to calculation of their $z$-scores. Note: $z$-scores are capped at –4.5 in Fig. 1C. Abbreviations: EO, eyes open; EC, eyes closed; WEO, weight eyes open; WEC, weight eyes closed.

self-reported upper limb function (DASH scores), (ii) ability to detect differences between 'off'/'on' medication states, (iv) feasibility with respect to test administration and (v) demonstrated reliability across the adult lifespan (*Ingram et al., 2019*). As indicated in Table 5, the handgrip strength, finger-press reaction time, 9-hole peg, bimanual pole, and arm stability tests along with the functional shirt task were identified as key tests.

## DISCUSSION

This study compared performance in a broad range of upper limb function tests between people with PD both with a healthy age- and gender-matched control group, as well as with normative values taken from people aged 50 years and over in the general population. These comparisons revealed that people with PD performed significantly worse across all domains with the exception of proximal muscle strength and tactile discrimination. Specifically, deficits were most apparent in fine motor control and dexterity, arm stability

**Table 3  Pearson product moment correlations ($r$) or Spearman rho correlations between individual test performance and age, duration of Parkinson's disease (Duration), Hoehn and Yahr (HY) score, and Disabilities of the Shoulder, Arm and Hand (DASH) questionnaire score during their 'off' and 'on' medication states.**

| Measure | 'Off' state | | | | 'On' state | | | |
|---|---|---|---|---|---|---|---|---|
| | Age | Duration | HY | DASH | Age | Duration | HY | DASH |
| Isometric elbow flexion strength (kg) | −.296 | .139[a] | −.321 | −.576** | −.438** | .201[a] | −.297 | −.531** |
| Handgrip strength (kg) | −.479** | .236[a] | −.325 | −.562** | −.517** | .232[a] | −.315 | −.549** |
| Finger-press reaction time (ms) | .132[a] | .199[a] | .643[a]*** | .470[a]** | .277 | .356[a]* | .579** | .510** |
| Finger tapping (no. of taps) | −.061[a] | −.114[a] | −.195[a] | −.052[a] | −.266 | −.056[a] | −.404* | −.169 |
| 9-hole peg test (s) | .175[a] | .294[a] | .476[a]** | .487[a]*** | .402[a]* | .297[a] | .529[a]*** | .396[a]* |
| Loop and wire test (no. of touches) | .222[a] | .290[a] | .425[a]* | .298[a] | .267[a] | .317[a] | .391[a]* | .100[a] |
| Position sense (°) | −.143[a] | .178[a] | .179[a] | .188[a] | .057[a] | .227[a] | .137[a] | .105[a] |
| Tactile sensitivity (g) | .232[a] | .202[a] | .212[a] | .171[a] | .328 | .385[a]* | .285 | .136 |
| Two-point discrimination (mm) | .440** | .192[a] | .375* | .552** | .295 | .115[a] | .315 | .251 |
| Two-line discrimination (mm) | .089[a] | .181[a] | .262[a] | .160[a] | .036[a] | .312[a] | .014[a] | .161[a] |
| Bimanual pole test (s) | .362[a]* | .286[a] | .407[a]* | .492[a]*** | .308[a] | .312[a] | .346[a]* | .448[a]*** |
| Arm stability – Eyes open (°) | −.026[a] | .185[a] | .271[a] | .261[a] | −.011[a] | .263[a] | .333[a] | .329[a] |
| Arm stability – Eyes closed (°) | −.087[a] | .227[a] | .244[a] | .213[a] | −.016[a] | .187[a] | .390[a]* | .373[a]* |
| Arm stability – Weight eyes open (°) | −.031[a] | .200[a] | .294[a] | .311[a] | .009[a] | .237[a] | .342[a]* | .301[a] |
| Arm stability – Weight eyes closed (°) | −.024[a] | .229[a] | .276[a] | .248[a] | .057[a] | .184[a] | .332[a] | .305[a] |
| Shirt task (s) | .299[a] | .082[a] | .539[a]*** | .374[a]* | .287[a] | .173[a] | .489[a]*** | .356[a]* |

**Notes.**

Negative values for isometric elbow flexion strength; handgrip strength and finger tapping indicate worsening performance with increasing age, duration, HY score and DASH score. Positive values for all other tests indicate worsening performance.

[a] Spearmans rho correlation reported for non-parametric variables.

*$p < 0.05$.

**$p < 0.01$.

***$p < 0.001$, uncorrected.

and functional tasks. When considered alongside correlations with the DASH questionnaire and test-retest reliability, these findings suggest that the handgrip strength, finger-press reaction time, 9-hole peg test, bimanual pole test, arm stability and the shirt task are key tests for assessing upper limb function in people with PD.

The results of the current study are largely consistent with that of previous research reporting impaired performance in single upper limb domains in people with PD when compared to the general population. Specifically, people with PD have deficits in measures of handgrip strength (*Roberts et al., 2015*; *Cano-de-la Cuerda et al., 2010*; *Koller & Kase, 1986*; *Villafañe et al., 2016*; *Jordan, Sagar & Cooper, 1992*; *Jones et al., 2017*), fine motor control (9-hole peg test) (*Earhart et al., 2028*; *Shah et al., 2019*), tactile sensitivity (when tested at the dorsum of the hand) (*Nolana et al., 2008*), simple reaction time (*Gauntlett-Gibson & Brown, 1998*), finger-tapping speed (*Jobbágy et al., 2005*; *Haaxma et al., 2010*; *Koop, Shivitz & Brontë-Stewart, 2008*; *Taylor Tavares et al., 2005*; *Lee et al., 2010*; *Yaholam et al., 2004*; *Memedi et al., 2013*; *Lee et al., 2016*; *Djurić-Jovičić et al., 2016*), position sense (during both active and passive elbow-matching tasks (*Zia, Cody & O'Boyle, 2000*), and movement detection thresholds (*Maschke et al., 2003*; *Konczak et al., 2007*), bimanual coordination (slower, less accurate and greater inter-trial variability) (*Ponsen et al., 2006*;

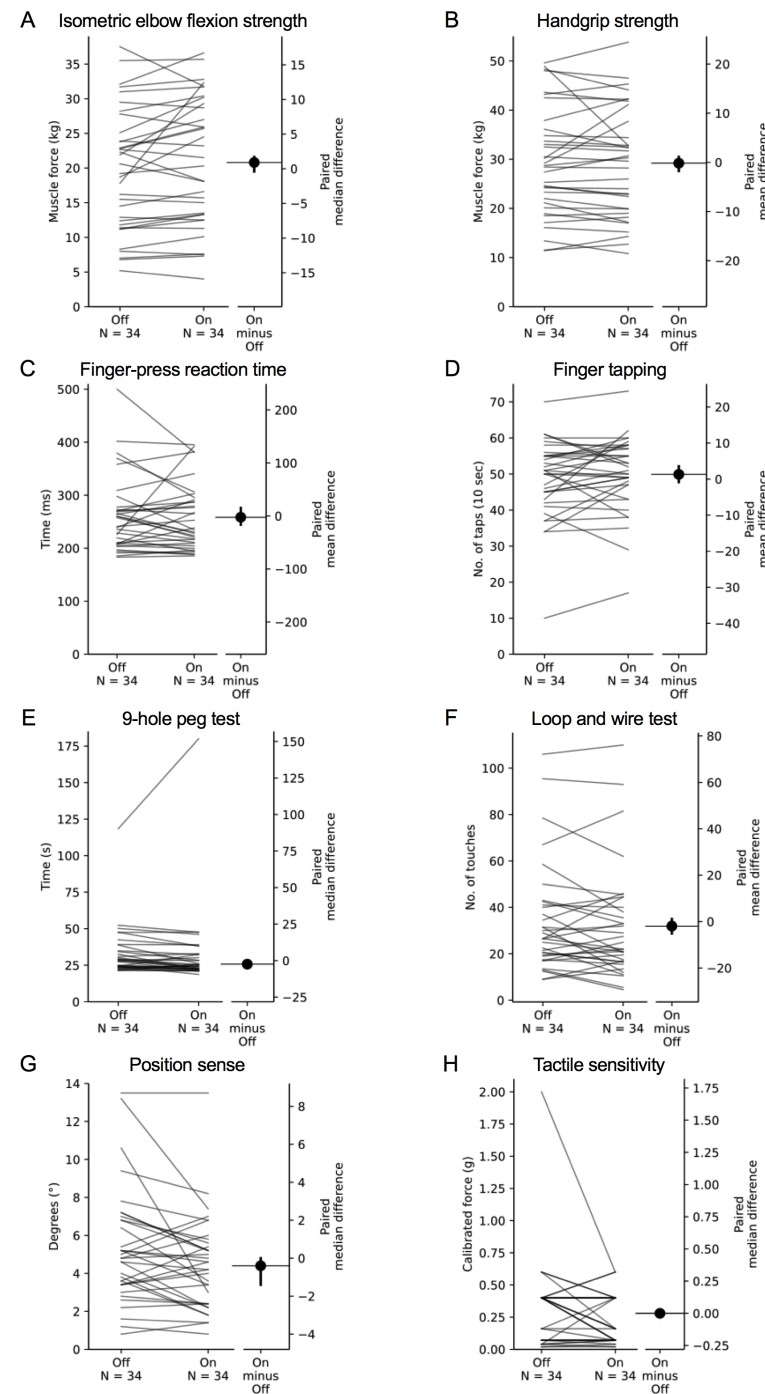

**Figure 7** **Comparisons of test scores in each test of the upper limb PPA between the 'off' and 'on' medication states in people with PD.** Each grey line plotted along the left axis of each graph represents the change in performance between the 'off' medication state (left-side of the line) and the 'on' medication state (right-side of the line) of each individual PD participant. The black circle located along the right axis of each graph represents the mean or median difference (depending on the distribution of the differences in test scores between medication states) in test scores between the 'off and 'on' medication states. Error bars depict 95% confidence intervals.

![PeerJ]

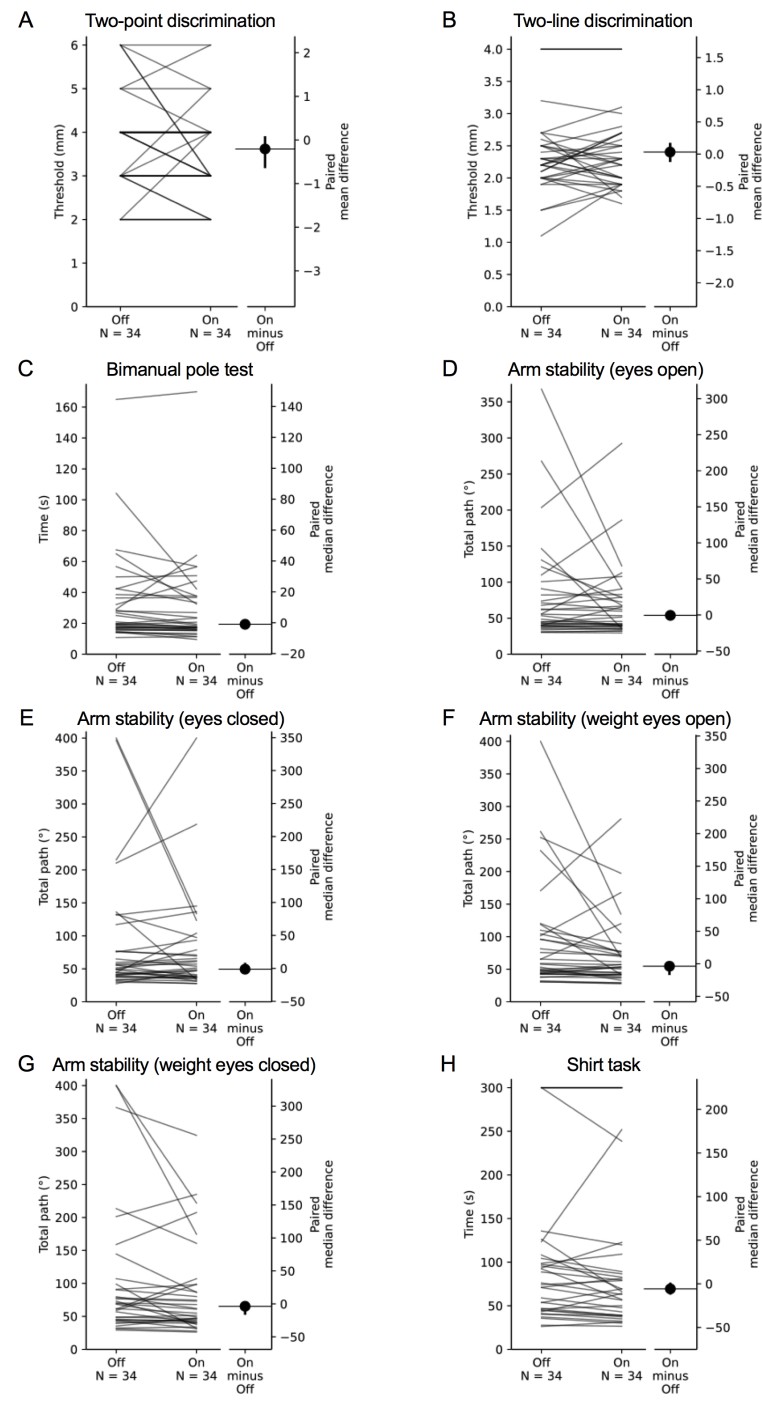

**Figure 8  Comparisons of test scores in each test of the upper limb PPA between the 'off' and 'on' medication states in people with PD.** Each grey line plotted along the left axis of each graph represents the change in performance between the 'off' medication state (left-side of the line) and the 'on' medication state (right-side of the line) of each individual PD participant. The black circle located along the right axis of each graph represents the mean or median difference (depending on the distribution of the differences in test scores between medication states) in test scores between the 'off and 'on' medication states. Error bars depict 95% confidence intervals.

**Table 4** Mean ± SD or median (IQR) scores for 'off' state and 'on' state for each test, mean or median difference [95% CI] in performance between 'off' and 'on' states (paired t-tests or Wilcoxon matched pairs test).

| Measure | 'Off' state | 'On' state | p | Difference |
|---|---|---|---|---|
| Isometric elbow flexion strength (kg) | 19.6 ± 8.8 | 20.9 ± 9.3 | 0.036[*] | 0.9 [−0.4, 1.7][a] |
| Handgrip strength (kg) | 29.4 ± 11.1 | 29.2 ± 11.1 | 0.813 | −0.2 [−1.8, 1.1] |
| Finger-press reaction time (ms) | 249.3 (208.0–274.1) | 236.5 (208.3–292.7) | 0.762 | −2.5 [−16.5, 15.2] |
| Finger tapping (no. of taps) | 50.5 (42.8–55.0) | 50.5 (46.0–57.3) | 0.252 | 1.3 [−0.9, 3.6] |
| 9-hole peg test (s) | 28.2 (24.1–35.9) | 25.0 (22.1–32.6) | 0.006[**] | −2.4 [−4.1, −0.6][a] |
| Loop and wire test (no. of touches) | 26.5 (18.6–41.4) | 23.5 (16.5–41.1) | 0.229 | −1.9 [−5.1, 1.1] |
| Position sense (°) | 4.8 (3.4–6.9) | 4.4 (2.4–5.7) | 0.017[*] | −0.4 [−1.4, −0.0][a] |
| Tactile sensitivity (g) | 0.28 (0.07–0.40) | 0.12 (0.07–0.40) | 0.072 | 0.00 [0.00, 0.00][a] |
| Two-point discrimination (mm) | 3.8 ± 1.1 | 3.6 ± 1.0 | 0.242 | −0.2 [−0.6, 0.1] |
| Two-line discrimination (mm) | 2.3 (2.0–2.5) | 2.3 (1.9–2.7) | 0.623 | 0.0 [−0.1, 0.2] |
| Bimanual pole test (s) | 20.4 (16.9–39.5) | 19.1 (16.0–37.6) | 0.032[*] | −1.0 [−3.0, −0.2][a] |
| Arm stability – Eyes open (°) | 54.5 (40.0–93.4) | 52.5 (38.6–80.1) | 0.614 | −0.5 [−2.6, 1.4][a] |
| Arm stability – Eyes closed (°) | 50.7 (39.5–86.9) | 52.6 (37.1–94.5) | 0.791 | −1.2 [−5.5, 6.4][a] |
| Arm stability – Weight eyes open (°) | 58.3 (43.2–105.6) | 55.4 (41.8–77.2) | 0.017[*] | −3.7 [−15.3, −1.6][a] |
| Arm stability – Weight eyes closed (°) | 68.9 (44.7–100.9) | 58.0 (40.9–98.3) | 0.023[*] | −3.6 [−14.7, −2.3][a] |
| Shirt task (s) | 75.1 (45.0–112.0) | 67.7 (39.0–111.8) | 0.060 | −5.8 [−10.9, 0.0][a] |

**Notes.**

Wilcoxon matched pairs test performed when distribution of differences between 'off and 'on states were non-parametric. Difference between on and off state reported as median [95% CI].

[*]$p < 0.05$.
[**]$p < 0.01$, uncorrected.

*Van den Berg et al., 2000*; *Almeida, Wishart & Lee, 2002*) and arm stability (*Barrantes et al., 2017*; *Lukšys, Jonaitis & Griškevičius, 2018*).

Comparisons of proximal upper limb strength between people with PD and healthy controls have been inconsistent, due in part to differences in the muscles tested (i.e., elbow extensors vs. flexors) and the parameters used to quantify strength. Specifically, people with PD demonstrate weakness in isometric elbow extension—not flexion (*Corcos et al., 1996*), and take longer to reach their peak torque during isometric elbow flexion and demonstrate more irregularities in their force-time profiles (*Stelmach & Worringham, 1988*; *Kunesch et al., 2005*). Our finding that tactile spatial acuity did not differ between PD and healthy control participants contrasts with a study by *Schneider, Diamond & Markham (1987)*, who reported a significant mean two-discrimination score of 4.6 mm at the index finger in a sample of 15 people with PD of comparable severity and gender distribution to that of recruited for the current study. The reason for these divergent findings is unclear, although may relate to differences in disease duration, not reported in the Schneider et al. study (*Schneider, Diamond & Markham, 1987*).

Our comparisons in performance between the 'on' and 'off' medication states contrast with some previous research. For example, *Corcos et al. (1996)* reported a 34% reduction in isometric elbow extension strength when comparing performance between 'on' and 'off' states, but no difference in elbow flexion strength. While this latter finding conflicts with

**Table 5  Validity, feasibility and reliability weightings for the upper limb PPA tests in people with Parkinson's disease.**

| Measure | <Healthy[a] | DASH[a] | L-dopa | Feasible | Reliable [20] | Total |
|---|---|---|---|---|---|---|
| Isometric elbow flexion strength | 0 | 1 | 1 | 0 | 1 | 3 |
| Handgrip strength | 1 | 1 | 0 | 1 | 1 | 4 |
| Finger-press reaction time | 1 | 1 | 0 | 1 | 1 | 4 |
| Finger tapping | 1 | 0 | 0 | 1 | 1 | 3 |
| 9-hole peg test | 1 | 1 | 1 | 1 | 1 | 5 |
| Loop and wire test | 1 | 0 | 0 | 0 | 1 | 2 |
| Position sense | 1 | 0 | 1 | 1 | 0 | 3 |
| Tactile sensitivity | 1 | 0 | 0 | 1 | 1 | 3 |
| Two-point discrimination | 0 | 0 | 0 | 1 | 1 | 2 |
| Two-line discrimination | 0 | 0 | 0 | 1 | 0 | 1 |
| Bimanual pole test | 0 | 1 | 1 | 1 | 1 | 4 |
| Arm stability – Eyes open | 1 | 0 | 0 | 1 | 1 | 3 |
| Arm stability – Eyes closed | 1 | 1 | 0 | 1 | 1 | 4 |
| Arm stability – Weight eyes open | 1 | 0 | 1 | 1 | 1 | 4 |
| Arm stability – Weight eyes closed | 1 | 0 | 1 | 1 | 1 | 4 |
| Shirt task | 1 | 1 | 0 | 1 | 1 | 4 |

**Notes.**

[a]Weightings calculated for PD group during on state < Healthy (Independent $t$-test or Mann–Whitney $U$ test $p$-values): $1 p < .05$, $0 p \geq .05$ (uncorrected).

DASH (Pearson correlation $r$-values): $1 — p < .05; 0 — p \geq .05$ (uncorrected).

L-dopa (Paired $t$-test $p$-values): $1 — p < .05; 0 — p \geq .05$ (uncorrected).

Reliability (test/re-test reliability): $1 — ICC \geq 0.7; 0 — ICC < 0.7$ or not calculable.

Feasibility: 1 minimal equipment required, short administration time. 0 requires bulky or insufficiently robust testing equipment.

ours, it is acknowledged the magnitude of difference in elbow flexion strength between medication states in our study was small (1 kg), and therefore of uncertain clinical relevance.

Other discrepancies are evident for position sense, matching tasks and reaction time. *O'Suilleabhain, Bullard & Dewey (2001)* reported that people with PD performed worse during their 'on' state during an elbow matching task which they attributed to levodopa-induced dyskinesias. The discrepancy between studies could be attributed to the O'Suilleabhain et al. study using a passive positioning test as opposed to the active positioning used in the current study, which would have potentially eliminated the role of central motor commands contributing to position sense at the elbow (for review, see (*Proske & Gandevia, 2012*; *Proske & Gandevia, 2018*)). While *Quincy & Brown (2013)* reported no improvement in bimanual task performance during the 'on' medication state, comparisons with the current study are limited due to differences in the test performed and the parameters used to quantify performance. Finally, differences in performances between medication states for measures of simple reaction time are equivocal (*Gauntlett-Gibson & Brown, 1998*; *Harrison, Henderson & Kennard, 1995*; *Velasco & Velasco, 1973*; *Bloxham, Dick & Moore, 1987*; *Puliman et al., 1988*; *Starkstein et al., 1989*; *Rafal et al., 1984*), with more complex tests (i.e., choice reaction time) appearing more sensitive in discriminating between the 'on' and 'off' states.

The standardised scores for both the 'on' and 'off' states indicate the extent to which people with PD are impaired in relation to normative values taken from people without

PD. The profiles scores indicate the PD participants performed markedly worse than the healthy controls in the reaction time, 9-hole peg test, loop and wire and arm stability tests ($z$-scores $<-1$) while being relatively less affected in terms of tactile sensitivity and muscle strength (with the exception of proximal strength in females during the 'off' state) (Fig. 5). These deficits are consistent with the cardinal symptoms of PD including bradykinesia, rigidity and tremor. In contrast, the relatively small finger tapping decrement, while consistent with a previous study by *Haaxma et al. (2010)*, was unexpected given this is an accepted marker of bradykinesia. Most notably, performance in the shirt task was distinctly compromised in people with PD in terms of time taken to complete the sequential task ($z$-scores $<-2$), which appears to reflect summed impairments in the physiological domains of dexterity, position sense, tactile sensitivity and bimanual coordination required for this functional task (*Ingram et al., 2019*). Indeed, people with PD have impaired performance in the specific tests that singly quantify each of these domains both within the current study and those previously reported—specifically, the 9-hole peg test (*Earhart et al., 2028*; *Shah et al., 2019*), elbow position-matching tasks (*Zia, Cody & O'Boyle, 2000*; *O'Suilleabhain, Bullard & Dewey, 2001*), von Frey filaments applied to the dorsum of the hand (*Nolana et al., 2008*), and both anti-phase (*Ponsen et al., 2006*; *Van den Berg et al., 2000*; *Almeida, Wishart & Lee, 2002*) and in-phase bimanual tasks (*Van den Berg et al., 2000*).

While performance in some tests correlated with the severity of PD symptoms—as measured by the HY scale, small to moderate associations with test performance were more consistent with a validated measure of upper limb function—the DASH questionnaire. This emphasises the importance of individual upper limb PPA tests for quantifying upper limb impairment and performance in functional tasks on the one hand, and the limited value of the HY scale in documenting upper limb disability in people with PD on the other (*Skorvanek et al., 2017*). This is not surprising given the HY scale's scoring criteria's emphasis towards impairments in postural stability and functional limitations in gait, especially towards the higher end of the scale. Despite being designed specifically for people with musculoskeletal shoulder disorders, the DASH appears sensitive in identifying upper limb impairments, activity limitations and participation restrictions in people with PD when compared to healthy control counterparts. An adapted version of the DASH questionnaire designed specifically for people with PD that incorporates the motor fluctuations commonly observed between the on/off states and quantifies the impact of key symptoms, such as tremor and bradykinesia, on everyday activities may refine the scale and provide further insight into the implications of PD from an activity and participation perspective (*Proud et al., 2015*).

Finally, in their 'on' compared with their 'off' medication state, the PD participants performed significantly better in tests of proximal muscle strength, finger dexterity, position sense, bimanual coordination and weighted arm stability. This suggests these tests would have value in distinguishing between those who do and do not respond to the administration of levodopa in clinical practice. Indeed, a computerised 25-peg insertion test—analogous to the 9-hole peg test, was sensitive to levodopa status in people with PD (*Müeller, Benz & Przuntek, 2002*). Importantly, this may be relevant to people with relatively early disease—such as the current sample (median time since diagnosis = 5.0

years), as people with PD generally first experience levodopa-related dyskinesias and motor fluctuations approximately 5–10 years after initiating levodopa treatment (*Bhidayasiri & Truong, 2008*). The lack of change in performance between 'off' and 'on' states for the shirt task is surprising. Perhaps compensation is most profound for complicated multimodal tasks.

Nevertheless, further studies with larger sample sizes are required to confirm the discriminant validity of the aforementioned tests in differentiating between medication states in the PD population. Furthermore, it must be cautioned that despite the statistical differences between medication states, the actual difference in scores between 'off' and 'on' states does indeed appear small in most cases, bringing the clinical significance of such findings into question. This is particularly true for position sense given its somewhat large coefficient of variation (CV) of measurement error when assessed in a healthy population (*Ingram et al., 2019*).

## Clinical implications

The study reveals the extent to which people with PD are impaired across the physiological domains necessary for adequate upper limb function: muscle strength, reaction time, fine motor control and dexterity, bimanual coordination, arm stability and functional tasks. Furthermore, as most of the tests demonstrate good criterion validity against the DASH questionnaire, and are simple and quick to administer, they constitute an excellent assessment battery for assessing upper limb for people with PD in the clinical setting. Here, they may play a valuable role in identifying specific motor impairments, monitoring disease progression and evaluating rehabilitation strategies designed to improve upper limb function. A number of the tests discriminate test performance between the "off" and "on" medication states, and offer scope for evaluating the efficacy of drug therapies and additional assessments to identify people with PD who do and do not respond to levodopa. The greatest asset that the upper limb PPA offers alongside the currently available tests is that it enables clinicians to quantify performance across multiple physiological domains necessary for independence of everyday activities with the upper extremities—each of which will be compromised to various degrees between patients with PD (*Jankovic, 2008*; *Politis et al., 2010*). This will allow the development of a performance profile that identifies the specific deficits (and their magnitude) in each individual, thereby tailoring rehabilitation and treatment to each patient's specific needs.

## Strengths and limitations

Strengths of the study included the comprehensive range of upper limb function tests, the assessments undertaken in both the on and off medication states, and the inclusion of participants across a broad age and PD severity ranges (1–4 on the HY scale). We also acknowledge some limitations. First, the Movement Disorder Society-sponsored revision of the Unified Parkinson's Disease Rating Scale (MDS-UPDRS) was not administered in our PD sample, so we are not able contrast upper limb PPA scores with this established PD severity rating scale. However, our analyses contrasted upper limb PPA performances with PD severity as assessed with the HY scale and it is important to note the upper limb PPA

was not designed as an alternative to the MDS-UPDRS scale, but rather a complementary quantitative battery of clinical tests that can identify specific sensory and motor impairments in individual patients. Second, due to logistical constraints, some participants may not have been in a complete 'off' state for their 'off' assessments, as it was not possible to withhold all PD medication for at least 12 h in all participants (*McKay, Harrigan & Brašić, 2019*). However, the majority of participants would have met this condition as they withheld their first daily PD medication dose till after their 'off' assessment. Therefore, if anything, our results may underestimate the ability of the upper limb PPA to document the effects of PD medication on upper limb function. For similar practical reasons, the assessor could not be blinded to medication status and the testing session order could not be randomised, and 27 of the 34 participants completed the initial session in their 'off' state and the second session in their 'on' state. Consequently, it is possible that learning effects may have influenced the 'on' assessments. This may be particularly the case for the bimanual pole test due to the novel nature of this task. Another limitation is that to minimize participant burden, participants performed the unilateral tests with the dominant hand only. However, given that 30 out of the 34 participants included in the current sample scored 2 or more on the HY scale (indicating bilateral involvement of PD-related symptoms), our results should still largely represent the sensorimotor impairments experienced by people with PD when performing everyday activities with their upper limbs. While we did not screen participants for cognitive impairment prior to participation, all included participants were able to follow the test instructions for each test without difficulty. Finally, although previous studies have reported good test-retest reliability for handgrip strength and the 9-hole peg test in people with PD (*Villafañe et al., 2016*; *Proud et al., 2019*) and we have reported acceptable test-rest reliability for all measures in a large sample of people without neurological conditions (*Ingram et al., 2019*), we did not assess test-retest reliability in the current study sample.

## CONCLUSION

In conclusion, the study findings reveal the extent to which people with PD are impaired across many physiological domains necessary for adequate upper limb function: muscle strength, unilateral movement and dexterity, bimanual coordination, arm stability and functional tasks. The upper limb PPA may therefore complement the MDS-UPDRS by providing individual upper limb performance profiles (as indicated in Fig. 6) to assist in understanding functional limitations and informing tailored treatments. As each of the tests that quantify performance in many domains demonstrate good validity when compared to the criterion of the DASH questionnaire, and are simple and quick to administer, they constitute an excellent assessment battery for assessing upper limb function in people with PD in the clinical setting. Here, they may play a role in identifying specific motor impairments, and evaluating pharmaceutical and physical rehabilitation therapies designed to monitor and address disease progression and improve upper limb function.

### Funding

This study was supported by the Motor Impairment Program grant from the National Health and Medical Research Council (NHMRC), #1055084. These funders provided salaries for authors (Annie Butler, Matthew Brodie, Simon Gandevia, Stephen Lord). The funders had no role in study design, data collection and analysis, decision to publish, or preparation of the manuscript.

### Grant Disclosures

The following grant information was disclosed by the authors:
Motor Impairment Program grant from the National Health and Medical Research Council (NHMRC): #1055084.

### Competing Interests

The authors declare there are no competing interests.

### Author Contributions

- Lewis A Ingram conceived and designed the experiments, performed the experiments, analyzed the data, prepared figures and/or tables, authored or reviewed drafts of the paper, and approved the final draft.
- Vincent K Carroll performed the experiments, authored or reviewed drafts of the paper, and approved the final draft.
- Annie A Butler analyzed the data, prepared figures and/or tables, authored or reviewed drafts of the paper, and approved the final draft.
- Matthew A Brodie analyzed the data, authored or reviewed drafts of the paper, and approved the final draft.
- Simon C. Gandevia and Stephen R. Lord conceived and designed the experiments, analyzed the data, authored or reviewed drafts of the paper, and approved the final draft.

### Human Ethics

The following information was supplied relating to ethical approvals (i.e., approving body and any reference numbers):

Ethical approval was granted by the Human Research Ethics Committee, University of New South Wales (HC 15607).

### Data Availability

Raw data is available in the Supplemental Files.

### Supplemental Information

Supplemental information for this article can be found online at http://dx.doi.org/10.7717/peerj.10735#supplemental-information.

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
