# Peer review of "Quantifying upper limb motor impairment in people with Parkinson’s disease: a physiological profiling approach"

_PeerJ, doi:10.7717/peerj.10735_

## Round 0.1 · original submission · Major Revisions

Both expert reviewers have provided careful thoughts and suggestions on your manuscript. Please make sure that your reply letter addresses each point.

Reviewer 1 ·

Basic reporting

Background literature should be increased concerning the measures of upper limb impairment in patients with PD. See, as an example, Proud et al., Arch Phys Med Rehabil, 2015.

Similarly, the obtained results should be compared to those already available in literature in the discussion section.

Experimental design

No comment

Validity of the findings

No comment

Additional comments

It is a well-conducted study on a crucial topic. The manuscript is clear and well written. I have just some comments that could improve the manuscript before publication.

1. My main concern is about the “key tests for assessing upper limb function in people with PD”. Was the weighting system already proposed in the literature? In that case, a reference should be provided. Otherwise, it should be clearly stated that the system was specifically proposed for this study. Moreover, I consider the use of reliability, as measured in a cohort of healthy subjects, incorrect. Reliability should be measured in the investigated cohort since it can be deeply influenced by the specific characteristics of the investigated sample.

2. The authors should describe in better detail why they decided to use two different samples as “control group”, i.e., the age- and gender-matched group, and the 50+ group, since it can be misleading for a reader.

3. As reported, patients were performed all the tests with their dominant hand. What about the differences between the most and the less affected side?

4. In the statistical analysis, considering the number of testing, did you apply a correction method?

5. It is unclear why the number of repetitions of each item was different. Muscle strength: the best of three trials in each test was used as the participant’s test score (line 90); Unilateral movement and dexterity: the average of two trials, one in each direction, was used as the participant’s test score (line 102); Position sense: with the participant’s score being the average of five trials. Skin sensation: three trials were performed in each direction, with the average of the average second and third trials calculated as the test score (line 123). Bimanual coordination, Arm stability, and Functional performance: unspecified, I suppose just one repetition. Please justify these differences among test repetitions.

6. Table: to improve clarity, I suggest merging Table 2 and Table 4.

7. Figures: to improve clarity, I suggest merging Figure 1 and Figure 3, as well as Figure 2 and Figure 4, reporting in each graph the following categories: “Off state”, “On state”, “Control”, “Control minus Off state”, “Control minus On state”,

8. Please amend the following line: lines 188 and 200 (Table 4.1 to Table 1); line 193 (spell out 27); line 194 (time information); line 332 (remove the first “in”).

Reviewer 2 ·

Basic reporting

1. As this is a validity study, I suggest you refer to the COSMIN checklist for guidance on the requirements for high quality studies of measurement properties. (https://www.cosmin.nl/wp-content/uploads/COSMIN-study-designing-checklist_final.pdf#). You are missing some important details. See below:
Define the type(s) of validity you are testing throughout the manuscript.
Lines 36 -40. Clearly state your hypotheses. What were you expecting?
It is also important to make clear in the aims that you have two separate questions – a comparison with selected age and sex matched controls and comparison with normative values.
Line 62. Give details of whether the PPA has been validated in people with PD
2. Introduction: Although you describe the cardinal impairments in PD, you need to provide a rationale for why this broad impairment-based assessment tool may be appropriate for the assessment of people with PD.
3. The reporting of results is difficult to follow particularly the section at Lines 213 and 226. Perhaps you need another heading at Line 213 as this is a different comparison and different question.
4. Results. In validity testing it is usual to place a greater emphasis on the strength of the correlations and less emphasis on statistical significance. Define and discuss further.
5. Discussion. Include more discussion of the findings in relation to other studies, and the clinical implications of the results. Strengthen your reasons as to why the PPA tests would be a valuable addition to the currently available tests.
6. You have inconsistencies in language throughout the manuscript. Also be careful in your use of terms like ‘function’.
e.g You have described the DASH as a measure of upper limb function in Line 62 and upper limb impairment in Line 202. Maintain accuracy and consistency. (Refer to Drummond, A. S., Sampaio, R. F., Mancini, M. C., Kirkwood, R. N., & Stamm, T. A. (2007). Linking the disabilities of arm, shoulder, and hand to the International Classification of Functioning, Disability, and Health. Journal of Hand Therapy, 20(4), 336-344)
3. Minor points:
P33-35. See comments below about defining ‘on’ and ‘off’.
Line 75. Check your wording. Reliability relates to the repeatability of a test.
Line 88. seated, not sat
Line 97. Use academic language ‘sychronised with’, not ‘synced to’ (informal language)
Line 154. Have you previously defined this abbreviation (IQR)?
Line 193. It is a writing convention to spell out numbers at the start of a sentence.

Experimental design

Line 47-50. Were there exclusion criteria relating to other upper limb neurological or musculoskeletal disorders which could affect test performance? If not, include this in the limitations of the study. Also did you include a cognitive test?
Line 52: Explain why you chose to match with two controls and describe how closely the participants and controls were matched in years.
Line 161-2: You introduce a new question and sample here. Are you making a comparison with published normative values? It isn’t clearly mentioned as one of your aims. Explain why you chose scores for people aged 50+ when the mean age of your participants was 68 years. Performance on many of these tests is influenced by age as shown in reference 19.
Line 167: ‘participants were unable to provide a test score’ – do you mean they were unable to complete these tests? Do you have a supporting reference for why you chose this method to obtain a score.?
Line 175. Explain why you chose not to use available participant scores which were available from the testing.
Line 194. As you mention in the limitations you have two different samples at different points in their PD medication cycle which may perform differently. Please justify why they are not treated as separate groups.
Line 313. I would challenge the notion that the test battery is quick to .
Reference 19 Table 5 indicates that there are statistically significant differences for some PPA tests, with women mostly scoring better. The PD sample has relatively fewer women and this may influence the results. I would suggest separate analysis for men and women for this question.
Line 353. impairments and functional limitations

Validity of the findings

Line 312. Define the criteria for acceptable validity. Are you sure you mean external validity here? Also Line 354 This is not external validity.
Line 221. Be careful about generalising your findings to the PD population. You can only know the results from this sample.
Line 288. The PPA tests do not document UL function, some tests are moderately associated with a questionnaire which assesses impairments, activity limitations and participation restrictions. Discuss the strength of these correlations. Are they similar to other studies? Why or why not?
P254 also Line 268. Ref 19 describes the reliability of the PPA in healthy adults. Reliability is specific to the individual population tested. Has reliability been evaluated in a PD sample? This is also relevant to Table 5
Line 278. Dyskinesia is also an important impairment. The figure shows that some participants had poorer performance in the on-phase arm stability test. Could this be associated with dyskinesia?
Line 282 Expand on this. Is there evidence from other studies that these deficits are present in people with PD?
Line 289. Discuss whether you would you expect this result given the basis of the HY.
Line 300 Need a bigger sample for this. Check studies of response to dexterity
Line 312. Acceptable external validity What do you mean here. Ext val refers to the
Table 4. While some of the score differences are statistically significant they are very small. Discuss in relation to measurement error.

Additional comments

Thank you for giving me the opportunity to review this manuscript. It is important to have valid measures for measuring the upper limbs in Parkinson’s disease.

---

## Round 0.2 · Minor Revisions

Thanks for addressing the reviewers' comments.

**Please address the following two points.**

Reviewer 2's comment about "participants were unable to provide a test score" meant they did not understand that wording. Please change the wording to something more standard, maybe "no score was available for X participants who could not complete the task".

About the multiple comparisons question, your point is well taken, but it would be appropriate to add ", uncorrected" to the line with p values in the notes under tables 2 through 5.

Reviewer 1 ·

Basic reporting

no comment

Experimental design

no comment

Validity of the findings

no comment

Additional comments

I recommend this paper for publication.

---

## Round 0.3 · accepted · Accept

Thank you for your patience with the longer-than-usual delay from submission to decision. I commend you for providing the raw data, which others may build on in the future.